# An Orthodontic Treatment Case of a TMD Patient with Maxillary Posterior Intrusion Using TSADs

**Gye-Hyeong Lee** [1,2]**, Jae Hyun Park** [3,4,*]**, Sang-Mi Lee** [2,5]**, Eun-Jeong Kim** [2,5]**, Seung-Weon Lim** [6] **and Danal Moon** [2,5]

1   Department of Orthodontics, College of Dentistry, Kyung Hee University, Seoul 02447, Republic of Korea
2   Department of Orthodontics, School of Dentistry, Chonnam National University, Gwangju 61186, Republic of Korea
3   Postgraduate Orthodontic Program, Arizona School of Dentistry & Oral Health, A.T. Still University, Mesa, AZ 85206, USA
4   Graduate School of Dentistry, Kyung Hee University, Seoul 02447, Republic of Korea
5   Department of Orthodontics, The Catholic University of Korea, Seoul 03312, Republic of Korea
6   Department of Dentistry/Orthodontics, Hanyang University Medical Hospital, College of Medicine, Hanyang University, Seoul 02447, Republic of Korea
*   Correspondence: jpark@atsu.edu; Tel.: +1-480-286-0455

**Abstract:** The orthodontic treatment of patients having temporomandibular disorders are some of the most complicated cases to treat. The positions of the mandibular condyles are often unstable, which means clinicians find it difficult to have definite criteria for making an accurate and reliable orthodontic diagnosis. This article reports the orthodontic treatment of a patient showing skeletal Class II and temporomandibular disorders with condylar resorption. To stabilize her condylar position and to relieve her symptoms in the temporomandibular joint, a stabilization splint was used before orthodontic tooth movement. After the splint therapy, the patient exhibited significantly increased open bite and a more severe Class II occlusal relationship as her mandibular condyles were seated anteriorly and superiorly in the articular fossae. The occlusion and facial esthetics of the patient were improved by orthodontic camouflage treatment with the proper use of temporary skeletal anchorage devices and treatment mechanics.

**Keywords:** temporomandibular disorders; orthodontic camouflage treatment; stabilization splint; TSADs; orthodontic miniscrews

## 1. Introduction

When using orthodontics to treat patients with temporomandibular disorders (TMDs), clinicians have found it difficult to establish a decisive treatment protocol. The etiologies are varied, and it is not easy to find the cause of TMDs [1,2]. When the etiology is due to problems with a patient's occlusion, there is a dramatic change in the occlusion after treatment of TMDs, as the position of the patient's condyles can be significantly altered. Therefore, the treatment of an orthodontic patient with TMDs should start with the stabilization of the condylar position, and a stabilization splint is approved to be the most reliable tool for stabilizing the surrounding musculoskeletal structures of a temporomandibular joint (TMJ) [3,4]. To record the occlusal relationship of a patient whose condyles are properly positioned in their most stabilized position and to fabricate the stabilization splint, it is essential to use an articulator [5,6].

After applying a stabilization splint in patients with TMDs caused by occlusal disorders, some show bite openings accompanied by premature contacts on their posterior teeth [7]. This can be caused by the clockwise rotation of the mandibular bodies, followed by a positional change of the mandibular condyles, generally to a more superior position in the articular fossae [8]. In the past, orthognathic surgery with maxillary posterior impaction was necessary to improve a patient's open bite when the intrusion of maxillary posterior

segments was required [9]. If orthognathic surgery could not be included in the treatment option, up-and-down elastics were alternatives for correcting anterior open bites, and the treatment effect was mainly due to the extrusion of the maxillary and mandibular anterior teeth. Consequently, these treatment mechanics can induce excessive exposure of the anterior teeth and an undesirable condylar positional change by providing a retraction force on the TMJ from the articular fossae.

With the use of temporary skeletal anchorage devices (TSADs), anchorage control is improved, and the efficiency of orthodontic treatment is better. Moreover, TSADs enable some types of tooth movement that were regarded as impossible with conventional orthodontic mechanotherapy. Although the mechanical behavior of TSADs could be affected by some variables like diameter and length of them, the crowning achievement of TSADs might be their ability to intrude the posterior teeth, which means the orthodontic treatment of anterior open bite with posterior early contacts can be resolved without orthognathic surgery or undesirable changes in the condylar position and esthetics [2,10]. This article presents an appropriate protocol and considerations of orthodontic treatment for patients with TMD.

## 2. Materials and Methods

*Case Report*

A 27-year-old female patient complained about her retruded chin position. She had suffered from jaw pain since her early teens and had started to have specific joint pain in her left TMJ area when she opened her mouth starting a few months ago. In the clinical examination, clicking sounds were heard on both of her TMJs when the patient opened her mouth. The patient had lip incompetency and showed a convex profile with facial asymmetry in which her chin point deviated to the right. An intraoral examination found moderate maxillary and mandibular crowding with an 11.5 mm overjet and an 0.5 mm anterior open bite. She had Class II canine and molar relationships on both sides, and her dental midline in the mandibular arch deviated 3 mm to the right. From the panoramic radiograph, the condylar heads on both sides of TMJ were found to be significantly flattened. The lateral cephalometric analysis indicated a skeletal Class II pattern (ANB = 11.5°) with a hyperdivergent growth pattern (SN-MP = 48.0°). However, the saddle angle was found to be in the normal range, and the mandibular body was sufficiently developed compared with the anterior cranial base length, while the ramus height was short compared with the posterior cranial base length, which suggested that any deformation of the TMJ structures might have had an influence on these skeletal features rather than the patient's sheer growth pattern (Figure 1; Table 1).

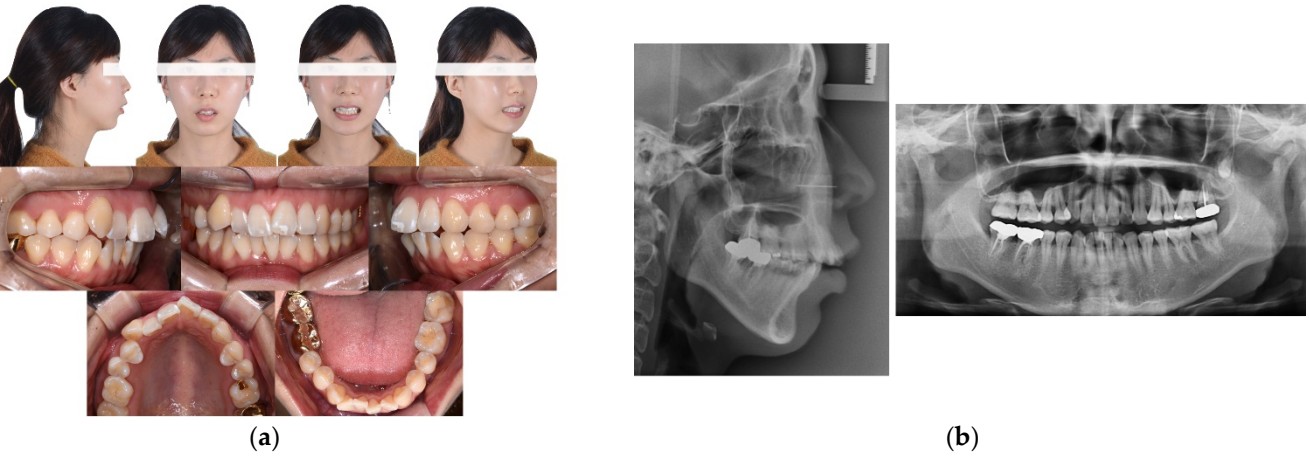

(**a**)                                        (**b**)

**Figure 1.** Pretreatment records: (**a**) facial and intraoral photographs, (**b**) lateral cephalogram, and panoramic radiograph.

**Table 1.** Cephalometric measurements.

| Measurement | Korean Norm | Pretreatment | Post-stabilization | Posttreatment |
|---|---|---|---|---|
| SNA (°) | 81.5 | 83.0 | 83.0 | 81.5 |
| SNB (°) | 79.0 | 71.5 | 70.5 | 72.0 |
| ANB (°) | 2.5 | 11.5 | 12.5 | 8.5 |
| Saddle Angle (°) | 123.0 | 122.0 | 120.5 | 121.0 |
| Articular Angle (°) | 149.0 | 167.0 | 169.0 | 168.5 |
| Gonial Angle (°) | 121.0 | 119.0 | 119.0 | 119.0 |
| SUM (°) | 393.0 | 408.0 | 408.5 | 408.5 |
| Anterior Cranial Base Length (mm) | 71.0 | 65.3 | 65.3 | 65.3 |
| Posterior Cranial Base Length (mm) | 33.0 | 36.0 | 35.0 | 35.0 |
| Mandibular Body Length (mm) | 71.0 | 72.0 | 72.0 | 72.0 |
| Ramus Height (mm) | 44.0 | 39.6 | 40.5 | 40.5 |
| ACB: MBL | 1:1 | 1:1.1 | 1:1.1 | 1:1.1 |
| PCB: Ramus Height | 3:4 | 3:3.3 | 3:3.5 | 3:3.5 |
| Post.FH/Ant.FH (%) | 66.8 | 58.0 | 57.3 | 57.3 |
| SN-MP (°) | 33.5 | 48.0 | 50.0 | 49.0 |
| TVL to MxOP (°) | 100.0 | 95.0 | 95.0 | 101.5 |
| U1-FH (°) | 116.6 | 115.0 | 115.0 | 100.0 |
| IMPA (°) | 90.0 | 90.0 | 90.0 | 92.0 |
| U1 to Stms (mm) | 3.0 | 3.9 | 3.9 | 3.0 |
| U1/L1 (°) | 124.0 | 99.0 | 95.0 | 119.0 |
| Overjet (mm) | 2.8 | 11.5 | 11.5 | 3.0 |
| Overbite (mm) | 3.0 | 0.5 | 0.0 | 2.8 |
| Upper lip (mm) | 0.0 | 4.3 | 3.8 | 0.0 |
| Lower lip (mm) | 0.0 | 5.9 | 5.0 | 0.0 |

*SNA*, sella–nasion–A point angle; *SNB*, sella–nasion–B point angle; *ANB*, A point–nasion–B point angle; *ACB*, Anterior craial base; *MBL*, mandibular body length; *PCB*, posterior cranial base; *FH*, facial height; *SN*, sella–nasion line; *MP*, mandibular plane; *TVL*, true vertical line; *MxOP*, maxillary occlusal plane; *IMPA*, incisor and mandibular plane angle; *Stms*, stomion superior.

Since the patient presented with signs and symptoms of TMDs, the position of the mandible needed to be evaluated, so her dental models were mounted on a semi-adjustable articulator (SAM III; SAM Praezision-stechnik GmbH, Munich, Germany) [10]. The mandibular position indicator (MPI) was applied to determine the reliability of the patient's jaw positions. MPI data was used to evaluate the centric-related occlusion (CRO)–maximal intercuspal position (MIP) discrepancies at the joint level. The MPI measurements indicated a 2.2 mm (left side) and 1.3 mm (right side) downward condylar distraction (Figure 2a).

Based on this diagnostic data, we concluded that the patient's mandible was in an unstable position, and therefore, her occlusion was not reliable enough to establish a definitive orthodontic diagnosis. To relieve pain that had been persistent in her TMJ area and to identify the stable condylar position of the mandible, a stabilization splint was suggested, an option which she accepted. A stabilization splint was fabricated on the SAM III articulator, and the patient was instructed in full-time use (Figure 2b). The splint was checked on a regular basis and adjusted to maintain a targeted occlusal scheme, a mutually protected occlusion. The stabilization splint played its role in eliminating the protective co-contraction of the surrounding masticatory muscles, and it led to an orthopedically stable joint position of the mandible. After a few weeks of the splint therapy, the pain in the patient's TMJ was reduced, but she was instructed to continue using the splint for weeks for the surrounding muscles to adapt to the TMJ structures. When we were convinced that her mandibular condyles were seated in their stable position based on MPI data and relieved TMJ symptoms, she was re-diagnosed for active orthodontic tooth movement. Her mandible was rotated clockwise as the condyles were seated into the most forward and uppermost positions, resulting in an increased anterior open bite (−1.5 mm) and a more severe skeletal Class II pattern, with a steeper mandibular plane (ANB, 12.5°; SN-MP, 50.0°) (Figure 2c,d; Table 1).

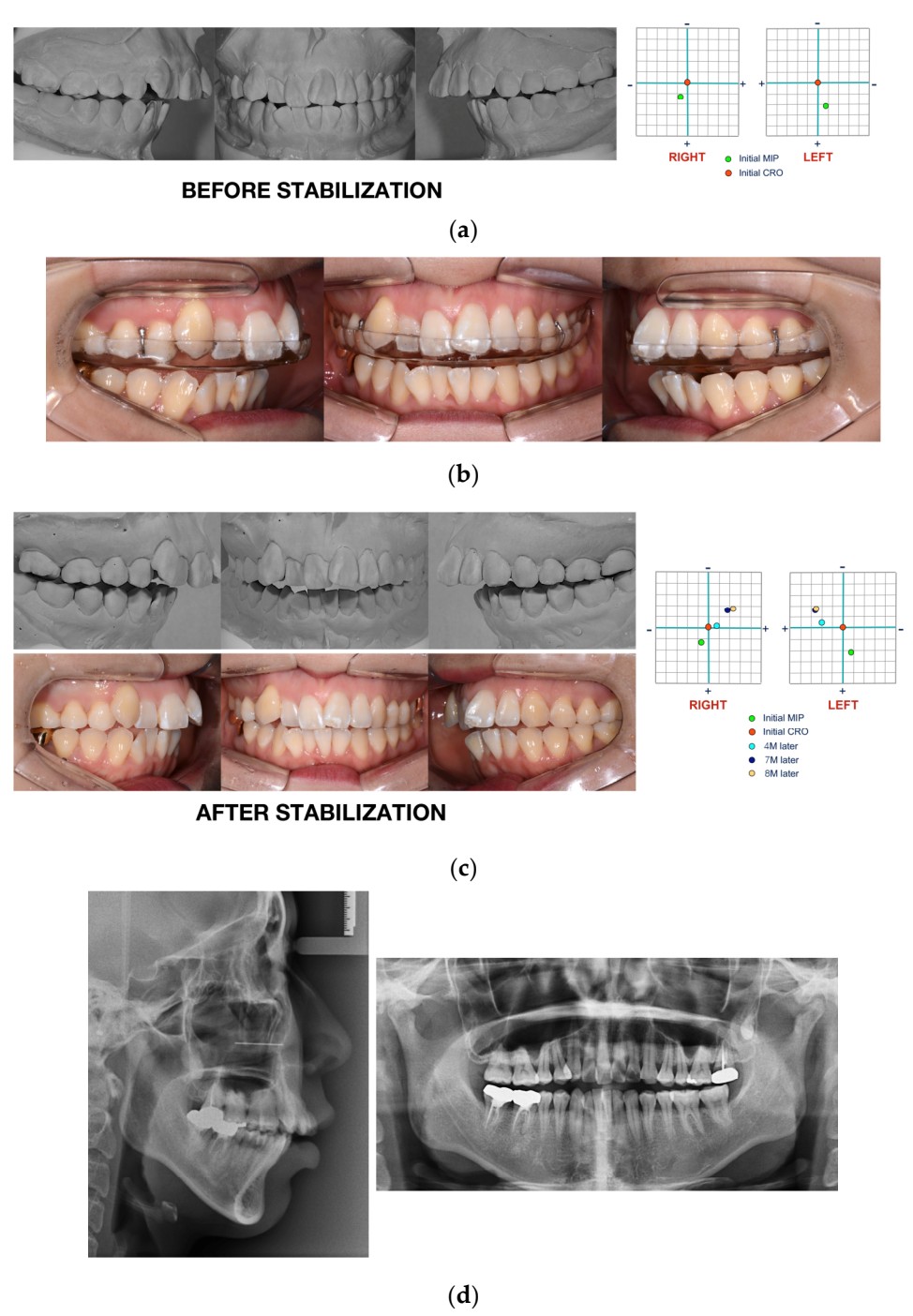

**Figure 2.** (**a**) Pretreatment dental models mounted in CRO and the MPI data. The MPI data presented MIP–CRO discrepancies at the joint level; (**b**) intraoral photographs during splint treatment; (**c**) post-stabilization intraoral photographs, dental models mounted in CRO, and changes of MPI data during the application of the stabilization splint. The MPI data discovered upward seating of the condyles in the articular fossae compared to their initial positions; (**d**) post-stabilization lateral cephalogram and panoramic radiograph.

With the re-diagnosed data, two treatment options were suggested to the patient. The first option included orthognathic surgery accompanying maxillary posterior impaction and mandibular advancement after presurgical orthodontic treatment with the extraction of the four first premolars. This treatment option effectively improved her open bite and facial esthetics. The other option was an orthodontic camouflage treatment with the extraction

of two maxillary first premolars and a left mandibular first premolar. This treatment alternative included the intrusion of her maxillary posterior teeth with the use of multiple TSADs to induce counterclockwise rotation of the mandible. These treatment mechanics were thought to eventually resolve the severe vertical and anteroposterior problems. The patient's maxillary anterior teeth had a normal inclination, while her overjet was 11.5 mm, which indicated that bodily movement of her maxillary anterior teeth would be needed during the retraction of the anterior segments, with her maxillary molars retaining absolute anchorage value. The left mandibular first premolar needed to be extracted to resolve the 7 mm arch length discrepancy and the denture midline discrepancy in her mandibular arch. After discussion, our patient rejected the invasive orthognathic surgery and chose the option of camouflage treatment using TSADs.

Before orthodontic treatment, both maxillary and left mandibular first premolars were extracted. Full fixed Roth prescription 0.018 in preadjusted edgewise orthodontic appliances (Tomy, Tokyo, Japan) were bonded on both arches for alignment and leveling. At six months of treatment, the retraction of the maxillary anterior teeth and intrusion of the maxillary posterior teeth was started. Two TSADs (diameter, 1.4 mm; length, 8 mm; Orlus, Seoul, Korea) were placed in the buccal alveolar bone between the roots of the maxillary first molars and second molars to intrude the posterior teeth and to retract the anterior segments. A transpalatal arch (TPA) was installed 5 to 6 mm away from the midpalatal soft tissue for the tongue exercise to intrude the posterior teeth more effectively. An additional TSAD between the roots of the maxillary central incisors beneath the anterior nasal spine (ANS) was implanted to control the root movement of the maxillary anterior teeth, in the expectation that this intrusive force would promote the bodily movement of the anterior teeth [11]. Two more TSADs were also placed in the posterior area of the plate for the intrusion of the maxillary molars. For the retraction of the anterior teeth, 100 g of orthodontic forces were applied on the elastomeric chains between each TSAD and $0.016 \times 0.022$-in stainless steel archwire. The same forces were also applied to the TPA and palatal TSADs for the intrusion of the posterior segments. These forces were reactivated and regulated until the proper overbite and overjet were achieved (Figures 3 and 4). The total treatment time was 29 months.

Posttreatment analysis discovered that both overbite and overjet were in normal ranges, and no TMD symptoms or discomfort was found after using the stabilization splint. A panoramic radiograph indicated no significant signs of progressed condylar resorption. The lateral cephalometric analysis presented minor skeletal changes (ANB, 11.5° to 8.5°), which were mostly induced by a counterclockwise rotation of the mandible, followed by the intrusion of the maxillary posterior dentition. On the superimposition of the lateral cephalograms, it was measured that there was a 3 mm maxillary molar intrusion with the use of TPA and TSADs. Inferring from the mandibular inferior border before and after treatment, it seems that the intrusion was symmetrical on both sides.

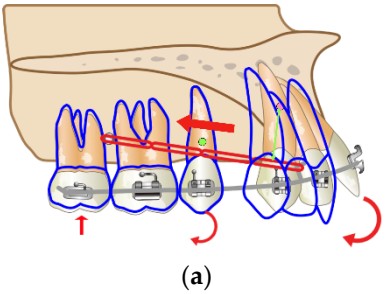    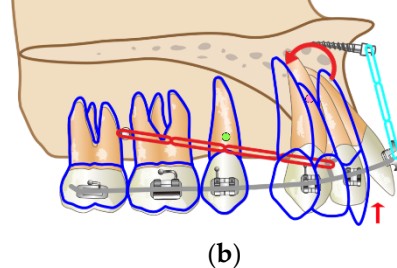

(a)                                              (b)

**Figure 3.** *Cont.*

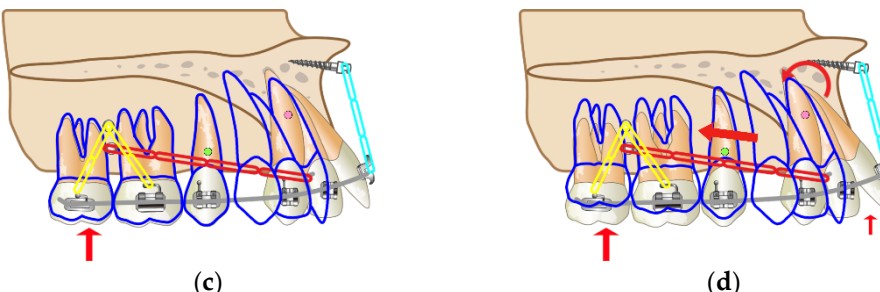

(**c**)　　　　　　　　　　　　(**d**)

**Figure 3.** Treatment mechanics: (**a**) TSADs were implanted between the maxillary first and second molars for absolute anchorage. During the retraction of the anterior segment, the moment is produced at the distance between the center of resistance (the point in the alveolus between the maxillary lateral incisors and canines) and the retraction forces for the anterior segment, which led to the clockwise rotation of the anterior teeth. Since the entire dentition was connected with a continuous heavy archwire, clockwise rotation of the full maxillary dentition occurred at the center of resistance, generally located 11 mm apical to and 26.5 mm posterior from the incisal edge of the maxillary central incisor, which resulted in the mild intrusion of the maxillary molars. (**b**) For additional root moment and palatal root movement of the anterior teeth, 100 g of distointrusive force passed through the center of resistance of the full maxillary arch with the help of a single TSAD installed beneath the ANS. (**c**) Additional TSADs for the intrusion of posterior teeth were implanted between the first and second molars on the palatal side. (**d**) The final treatment mechanics.

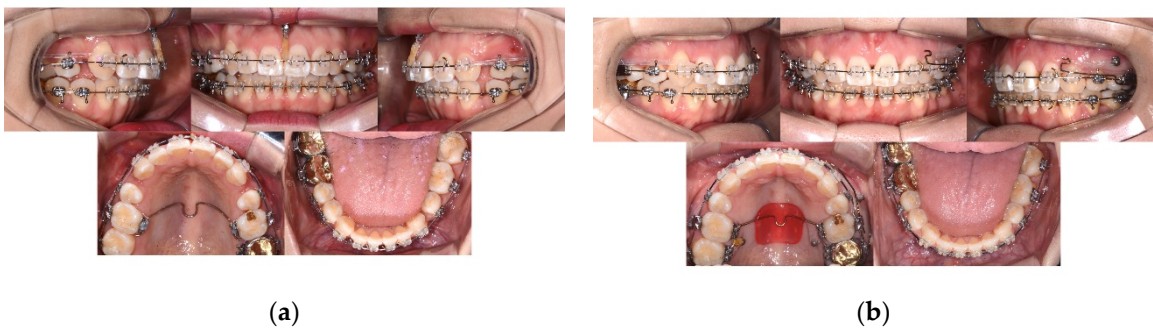

(**a**)　　　　　　　　　　　　(**b**)

**Figure 4.** Treatment progress: (**a**) a single TSAD installed beneath the ANS and two buccal TSADs between the molars were used for the retraction and intrusion of anterior teeth. (**b**) Intrusion force was applied to the maxillary posterior teeth with a combination of TPA and maxillary buccal TSADs.

Moreover, the intrusion of the posterior teeth provided the patient with a more esthetic smile line, as her flat maxillary occlusal plane angle was improved. To reduce overjet, her maxillary anterior teeth were retroclined (U1-FH, 100.0°). These changes contributed not only to the correction of the anterior open bite but to the improvements in the facial profile (Figure 5; Table 1).

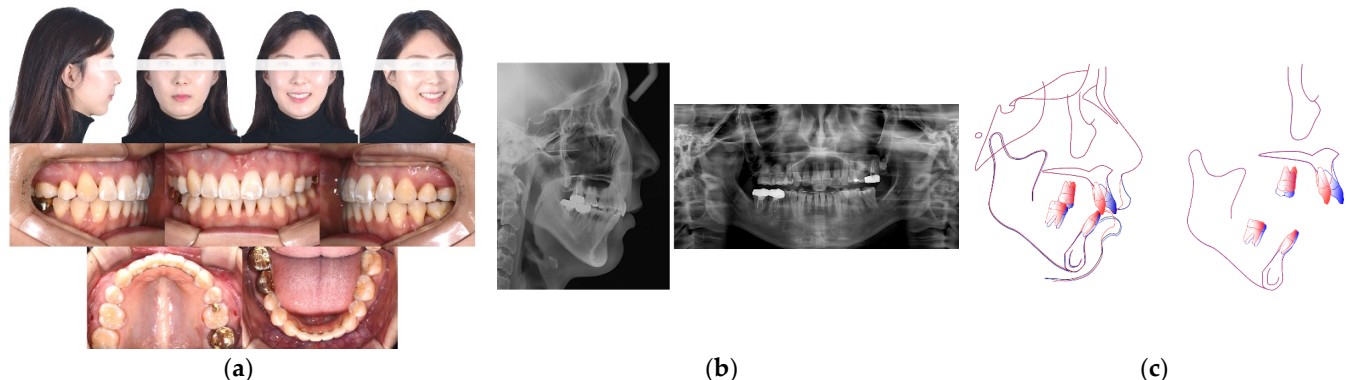

**Figure 5.** Posttreatment records: (**a**) facial and intraoral photographs, (**b**) lateral cephalogram and panoramic radiograph, and (**c**) cephalometric superimposition. Black, pretreatment; blue, post-stabilization; red, posttreatment.

## 3. Discussion

### 3.1. Importance of Condylar Position

Although there are controversies about occlusion as an etiology of TMDs, providing the patient with a stable and well-functioning masticatory system is as important as the establishment of dental and facial esthetics and intercuspal stability in orthodontic treatment [11]. In support of this, Okeson has claimed that it is important to achieve orthopedically stable joint positions that contribute to a stable occlusal position [12].

The definition of the proper condylar position has been discussed among researchers for decades, and most recently, this position is being called centric relation (CR). CR refers to "a maxillomandibular relationship, independent of tooth contact, in which the condyles articulate in an anterosuperior position against the posterior slopes of the articular eminences". The mandible is restricted to a pure rotation movement in CR, and the patient can make vertical, lateral, or protrusive movements that take a clinically beneficial and repeatable reference position [13]. The occlusion of opposing teeth is defined as centric relation occlusion (CRO) when the mandible is in CR. Maximum intercuspal position (MIP) is defined as "the most closed position that the mandible can assume, determined by full intercuspation of the opposing teeth, regardless of condylar position". If occlusal interference occurs during the closing movement of the jaw, opening muscles such as inferior lateral pterygoids contract to protect the interfering teeth, which is called "protective co-contraction" [6]. The consistent repetition of the proprioceptive stimulation to the masticatory muscles can promote a deviated closure pattern of the mandible, and the displacement of the condyle position in the MIP can be derived from these patterns of muscle activity. This positional difference of the condyle is known as the MIP–CRO discrepancy [14–17].

Numerous electromyographic studies have shown that the occlusal interference disrupts the coordination of the function in the masticatory muscle [18–23]. TMDs can be aroused when a hyperactivation of the muscles produced by protective co-contraction exceeds the patient's musculoskeletal adaptability. Cordray's study reported that almost all symptomatic subjects of TMD (96.9%) exhibited significant condylar displacement in at least one plane [4]. Crawford also revealed an 84% reduction in TMD signs and symptoms after full mouth restoration treatment, resulting in almost no MIP–CRO discrepancy compared to a non-treatment control group. Furthermore, he concluded that "since the condylar position is dictated upon closure of dentition into maximum intercuspation and since the condylar position was shown to be strongly correlated with TMD symptomatology, a statistically significant relationship existed between occlusion-dictated condylar position and symptoms of TMD" [15].

Clinicians have often measured the amount and direction of MIP–CRO discrepancy chairside by making purely intraoral observations. However, these chairside observations

lack analytical precision as the patient's mandible is not being guided correctly due to protective neuromuscular reflexes [15–17,24]. Another common method to measure MIP–CRO discrepancies is to verify the amount and direction of hit-and-slides at the occlusal level, but some researchers have found that the amount of slides at the occlusal level did not coincide with the 3-dimensional change at the condylar axis level [15,16,24]. Comparing the position of the condyle in radiographic images as panoramic and TMJ views is also inadequate due to the distortion of the 2-dimensional images [24]. The most accurate and reliable method to measure the amount and direction of the MIP–CRO discrepancy in three dimensions is to use articulator-mounted study models in the CRO in conjunction with condylar position instrumentation [24–26]. The amount and direction of the MIP–CRO discrepancy can be measured by tracing the changes in the position of the condylar hinge axis in the three planes: horizontal, vertical, and transverse planes in the condylar position indicator of the articulator. The accuracy, reproducibility, and reliability of this method have been confirmed, and it is easy to apply, is non-invasive, and costs less [27].

To treat orthodontic patients with TMD, it is important to evaluate whether the TMD symptoms are due to occlusion factors. They often show a protective neuromuscular reflex, which makes it more difficult to find the point of deflective occlusal contacts [14,28]. The only method to confirm the occlusion as an etiology is to apply a stabilization splint to the TMD patients [28–31]. By using a stabilization splint, clinicians can identify the occlusal interferences and uncover the true maxillomandibular relationships that were previously unrevealed by the protection of the patient's neuromuscular system. If the etiology comes from an occlusal factor, the TMD symptoms will be alleviated after the appropriate use of a stabilization splint. The stabilization splint is constructed in centric relation, which means the patient's condyles should be seated in their most stable position in the articular fossae. All mandibular buccal cusps and incisal edges should make contact on the flat surfaces of the splint with even force and with the proper inclination for anterior guidance, allowing the immediate disclusion of posterior teeth in protrusive or lateral movements [32]. When the stabilization splint functions in an ideal occlusal scheme, it reduces abnormal muscle activity by eliminating the protective co-contraction of the surrounding masticatory muscles. Furthermore, it also elongates elevator muscles, which induces neuromuscular relaxation and reduces pain around the temporomandibular structures [33,34]. The use of a stabilization splint also promotes the formation of a "pseudodisc" on the posterior band of the disc. It helps the condyle seat in the physiologically functioning position in the articular fossae [35–37].

### 3.2. Considerations for Orthodontic Tooth Movement after Stabilization

Williamson et al. reported that splint therapy in TMD patients resulted in increased open bite as condyles moved in the anterosuperior direction of the articular fossae [38]. Girardot et al. studied 19 patients with TMD symptoms who had stabilization splints applied full-time until the symptoms were relieved and discovered that their condyles moved superiorly from the condylar position, indicating instrumentation data [24]. Park et al. also claimed in their report that if an occlusal interference persists, a protective co-contraction in the masticatory system can occur, and positional changes of the mandibular condyles can be produced, which are generally caused by the uncontrolled contraction of the lower lateral pterygoid muscles and the hyperactivation of the surrounding elevator muscles [6]. When etiologic factors are resolved by a stabilization splint, and the harmful protective co-contraction is eliminated, the mandibular condyles gradually move superoanteriorly in the articular fossae [1,34,39]. This position is naturally guided by the vector sum of the three major elevator muscles: masseter, temporalis, and medial pterygoid muscles [2,6,9,14].

If the condyles move superiorly in the articular fossae after a stabilization splint has been used, the posterior teeth act as fulcrum points that produce a clockwise rotation of the mandible and cause a bite opening and/or more severe Class II tendency. Lim et al. reported that, after the splint therapy, orthodontic patients with large MIP–CRO discrepancies showed a greater difference in ANB, a larger overjet, and a steeper mandibular

plane [5]. In some cases, these patients require orthognathic surgery due to the dramatic occlusal changes, such as severe open bite and a retrognathic mandible. The vulnerable joint structures in TMD patients, however, may not be strong enough to tolerate the mechanical loads produced after osteotomies for mandibular advancement. If the mechanical loads to the TMJ structures exceed the biological adaptability, unexpected condylar resorptions or the progressive positional change of the condyles could also occur. Therefore, in some cases, clinicians might have to consider using orthodontic camouflage treatment [2,40–42].

In the orthodontic camouflage treatment of patients showing anterior open bite and Class II occlusal relationship, two requisite considerations should be kept in mind: the maxillary anterior teeth exposure and the inclination of the maxillary occlusal plane [1]. The appropriate exposure of the maxillary anterior teeth is closely related to smile esthetics [43]. If the exposure of the maxillary anterior teeth is excessive, the patient will present an unesthetic gummy smile, while if it is not sufficient, the patient will look older. The inclination of the maxillary occlusal plane influences both on the function of mastication and facial esthetics [44,45]. If a patient has a steep maxillary occlusal plane, the chances that occlusal interference occurs during the mandibular movements will increase [46]. When the point of early contact on the mandibular teeth acts as a fulcrum, the displacement of the mandibular condyles can result. The anteroposterior chin position is also affected by the inclination of the maxillary occlusal plane. The more the maxillary occlusal plane is steepened, the more convex the facial profile is presented. In contrast, the more the maxillary occlusal plane is flattened, the more pronounced the chin point is presented [47]. Choi et al. reported in their investigation that the vertical position of the maxillary posterior teeth should be positioned such that the inclination of the maxillary occlusal plane to the patient's true vertical line is 100° and claimed that, with this inclination, the occlusal function and esthetics could be optimized [48].

The use of intermaxillary elastics to correct a Class II open bite could be detrimental to TMD patients. Gurbanov et al. reported from their finite element study that the stresses in both condyles and the disc were greater in Class II patients than in Class III patients when orthodontic elastics were applied, and when the disc was in an anterior position, the stresses were likely to be more injurious to the retrodiscal tissue [49].

In the treatment of patients showing a anterior open bite and a Class II skeletal and dental relationship, many studies have reported successful orthodontic camouflage treatment using TSADs to improve not only occlusions but also facial esthetics [50–54]. In some patients, the intrusion of the posterior teeth using TSADs promoted profile improvement nearly similar to those of surgically assisted maxillary posterior impaction. This treatment alternative could result in the counterclockwise rotation of the mandible and could be a possible choice of treatment to improve the chin prominence and reduce the anterior open bite and anterior facial height [54]. It also has an advantage for the stability of the condylar position, as it does not generate an injurious force in the direction that causes condylar distraction [1,2,9,55]. Though there will probably be a bit of relapse in the intrusion of the maxillary posterior teeth using TSADs, the mandibular molars may show slight extrusion as the maxillary posterior teeth are being intruded [51,53]. The mild extrusive relapse of the maxillary posterior teeth and the simultaneous eruption of the mandibular posterior teeth might decrease the effect of the counterclockwise rotation of the mandible. Therefore, controlling the vertical position of the mandibular posterior teeth assumed great importance in increasing chin prominence and improving dental relationships. Long-term stability after the treatment of a severe Class II relationship or anterior open bite using TSADs has not been fully investigated, but many researchers have proven the short- and long-term stability of this treatment strategy [56–58].

## 4. Conclusions

When orthodontically treating TMD patients, an evaluation of the positional stability of mandibular condyles is critical. If the patient's condyles are not properly positioned and their functions are not stable, the application of a stabilization splint should be the

first step in the treatment process. The final orthodontic treatment plan needs to be based on the stabilized skeletal and occlusal relationship, and the treatment mechanics should ensure that the condyles remain in their stabilized position during the orthodontic tooth movements. Some patients who reveal an increased anterior open bite and a more severe Class II occlusal relationship after the stabilization of the condyles can be treated with posterior intrusion with the proper use of TSADs.

**Author Contributions:** Conceptualization, G.-H.L. and J.H.P.; writing—original draft preparation, G.-H.L., S.-M.L., E.-J.K., S.-W.L. and D.M.; review and editing, J.H.P.; treating the case, D.M. All authors have read and agreed to the published version of the manuscript.

**Funding:** This research received no external funding.

**Informed Consent Statement:** Informed consent was obtained from the subject involved in the study. Written informed consent has been obtained from the patient to publish this paper.

**Data Availability Statement:** The data presented in this study are available upon request from the corresponding author.

**Conflicts of Interest:** The authors declare no conflict of interest.

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
