# Peer review of "An Orthodontic Treatment Case of a TMD Patient with Maxillary Posterior Intrusion Using TSADs"

_applsci, doi:10.3390/app122312098_

Round 1

Reviewer 1 Report

This is not review, this is case report. 

In the title it is not clear whether the patient have posterior intrusion?

The photos should be of much better quality and resolution. Also the identity of the patient should be hidden.

In table the abbreviations for variables should be explained. 

In the discussion section, the authors should also discuss the results obtained. Since in this case the counterclockwise rotation of the mandible did not happen.

Reviewer 2 Report

The article presented to me for evaluation deals with a difficult clinical problem, which is the treatment of patients with TMD and Class II occlusal relationship . From this point of view, the article is valuable and will certainly be of interest to orthodontics and maxillofacial surgery specialists.

This paper is definitely a case study, not a review article. Therefore, it should be included under the case report category, even though the discussion is based on numerous references.

Below are some detailed considerations that should be taken into account when drafting the final manuscript:

1. During the treatment, only the intrusion of the upper molars was performed, which should be clearly specified in the title of the work. The title should also contain information that it is a Case Report.

2. There is no information in the text about the distance by which the maxillary molars were intruded. A measurement, for example the distance from vestibular mesial cusp of a molar to the palatal plane, should be given. It would also be important to know whether the intrusion was symmetrical on both sides. It would also be interesting for readers to indicate whether the pure molar intrusion was obtained through the use of TPA and TSADs, or whether it was an intrusion combined with an vestibuloangulation of the molar crowns. This assessment can be made on a CBCT or PA cephalometric x-rays.

3. There is no literature reference for the statement “In the past, orthognathic surgery with maxillary posterior impaction was necessary to improve a patient's open bite when intrusion of maxillary posterior segments was required” (lines 46-48) - please give a reference.

Overall, this case report is interesting and should be published after addressing the points listed above.

Reviewer 3 Report

The manuscript is well written and the topic is interesting 

but I think that the authors must to specify because they 

have not used the digital work flow (intraoral scanner

3D printing) that today are the standard in the clinical practice.

Author Response

Thank you for your sincere advice.

Unfortunately, this case was not treated with digital work flow.

If I had another chance to submit an article, I would prepare some case report with digital work flow.

Round 2

Reviewer 2 Report

I am satisfied with the amendments made by the authors.

Author Response

Thank you.